# Unmasking a gap: A new oligoneuriid fossil (Ephemeroptera: Insecta) from the Crato Formation (upper Aptian), Araripe Basin, NE Brazil, with comments on *Colocrus* McCafferty

**Arianny P. Storari** [1]*, **Taissa Rodrigues**[1], **Antonio A. F. Saraiva**[2], **Frederico F. Salles**[3]

1 Laboratório de Paleontologia, Departamento de Ciências Biológicas, Centro de Ciências Humanas e Naturais, Universidade Federal do Espírito Santo, Vitória, ES, Brazil, 2 Laboratório de Paleontologia da URCA—LPU, Centro de Ciências Biológicas e da Saúde, Universidade Regional do Cariri, Crato, CE, Brazil, 3 Museu de Entomologia, Departamento de Entomologia, Universidade Federal de Viçosa, Viçosa, MG, Brazil

* ariannystorari@gmail.com

**Data Availability Statement:** All relevant data are within the paper and its Supporting Information files.

## Abstract

The Crato Formation (Araripe Basin) preserves one of the most diverse entomofaunas of the Cretaceous. Among the groups of insects, mayflies stand out in abundance, but among them oligoneuriids are especially rare. A newly discovered adult oligoneuriid from this unit is here described as *Incogemina nubila* gen. et sp. nov. and new subfamily Incogemininae. A phylogenetic analysis recovered the new taxon as the sister group to the species-rich and cosmopolitan Oligoneuriinae. The paratype of *Colocrus indivicum*, described as an "oligoneuriid" from the same unit, is here reviewed and considered as belonging to the family Hexagenitidae. The biogeographical and taxonomic implications of this discovery and the phylogenetic position of *Incogemina nubila* are discussed. *Incogemina* bridge an important morphological gap between the Oligoneuriinae and the extant *Chromarcys*. Also, it demonstrates that the divergence between Oligoneuriinae and Incogemininae probably occurred in South America.

## Introduction

The Crato Formation (Araripe Basin) in northeast Brazil is a geological unit that preserves one of the most diverse entomofaunas of the Cretaceous [1]. Mayfly larvae constituted important elements of this fauna and, as they had fully aquatic lifestyles, they are more prone to be preserved as fossils than the alates (imagoes or subimagoes). This preservational bias is reflected in the strong disparity in the number of fossilized adult individuals in scientific collections, when compared to the much higher number of larvae, e.g. in McCafferty's work [2] the larvae to alate specimens ratio is approximately 4:1. However, in a controlled excavation in the Crato Formation led by one of us (AAFS), the ratio between larvae and adults was even more outstanding, since no adults were recovered, in contrast to 151 larvae. Regarding the Crato Formation, there are five named mayfly species described based on adult type material, and eleven based on larvae [3].

**Funding:** This study was financed in part by the Coordenação de Aperfeiçoamento de Pessoal de Nível Superior - Brasil (https://capes.gov.br/) - Finance Code 001, Conselho Nacional de Desenvolvimento Científico e Tecnológico (http://cnpq.br/) - 309666/2019-8 to TR and 309666/2019-8 to FFS, and by the Fundação Cearense de Apoio ao Desenvolvimento Científico e Tecnológico (https://funcap.ce.gov.br/) to AAFS (#BP3-013900202.01.00/18). The funders had no role in study design, data collection and analysis, decision to publish, or preparation of the manuscript.

**Competing interests:** The authors have declared that no competing interests exist.

Among fossil mayflies, oligoneuriids (Oligoneuriidae) are especially rare. The family is divided into three subfamilies [2, 4]: Colocrurinae McCafferty, 1990; Chromarcyinae Demoulin, 1953; and Oligoneuriinae Ulmer, 1914. Colocrurinae is a subfamily known only by fossils and comprises two species of the genus *Colocrus* McCafferty, 1990, both from the Crato Formation (*Colocrus indivicum* McCafferty, 1990 and *Colocrus magnum* Staniczek, 2007). Chromarcyinae is monotypic, represented by the extant species *Chromarcys magnifica* Navás, 1932, with an Indomalayan distribution [5]. All remaining 67 species and ten genera are extant and included in the Oligoneuriinae, and exhibit a predominantly Pantropical distribution [4, 6].

A newly discovered adult individual from the Crato Formation is here described and identified as a new genus and species. We also review the adult paratype of *Colocrus indivicum*, which is here considered as belonging to an undetermined genus of the family Hexagenitidae.

## Institutional abbreviations

AMNH, Invertebrate Zoology Collection of the American Museum of Natural History, New York, USA

LPU, Paleontology Collection of the Regional University of Cariri (URCA), Crato, Brazil

SMNS, Staatliches Museum für Naturkunde, Stuttgart, Germany

## Material and methods

The specimen LPU 1696 was collected in an outcrop of the Crato Formation at the Mine Antônio Finelon (S 07° 07' 22,5" and W 39° 42' 01") in Nova Olinda municipality, Ceará State, Brazil (Fig 1). The material was recovered from the top level carbonate C6 [7]. More detailed geological and sedimentological information about the Crato Formation can be found in Martill et al. [8].

This new specimen (Fig 2) was analyzed using a Leica binocular microscope. All drawings were made with a Wacom tablet, using the software Autodesk Version 8.6.1, and the photos were taken with a Nikon D800 digital camera. Pictures using ultraviolet light were taken using a Canon EOS Rebel T6i camera. The descriptive anatomical terminology is based on Kukalova-Peck [9] and Kluge [10]. Pictures of AMNH 43499 (Fig 3), the adult paratype of *Colocrus indivicum*, were made available by the American Museum of Natural History. The map (Fig 1) was generated using the QGIS software, version 3.10 [11], with shapefiles provided by the Serviço Geológico do Brasil (CPRM–GeoSGB).

In order to determine the phylogenetic relationships of LPU 1696, a cladistic analysis was performed using morphological data from the Oligoneuriidae matrix presented by Massariol et al. [4]. Three new characters were proposed, and characters 32 (insertion of gill I) and 53 (orientation of vein RP2 of forewing in relation to RA) were recoded in the matrix according to our interpretation (see S1 and S2 Appendices). The venational nomenclature was revised after Kukalova-Peck [9]. The combined data analysis included both larval and adult characters, with 76 binary characters in total. We also added LPU 1696 to their matrix, thus analyzing 19 ingroup taxa, representing all 12 oligoneuriid genera. We have excluded from the analysis the undetermined species *Lachlania* sp. and *Homoeoneuria* sp. because their genera were already represented by other species, and the intention of the analysis was to unravel the phylogenetic position of LPU 1696 within Oligoneuriidae. Outgroup taxa were the same used by Massariol et al. [4], from the related families Isonychiidae, Coloburiscidae, and Heptageniidae, which together with Oligoneuriidae compose the superfamily Heptagenioidea [10, 12].

Parsimony methods were conducted using TNT 1.5 [13] (data matrix in S2 Appendix). All characters were treated as non-additive and unordered. An exhaustive search was run under the implicit enumeration command, and implied weights, testing several concavity constant values (k = 1–20). The implied weights were used because they normally increase the stability

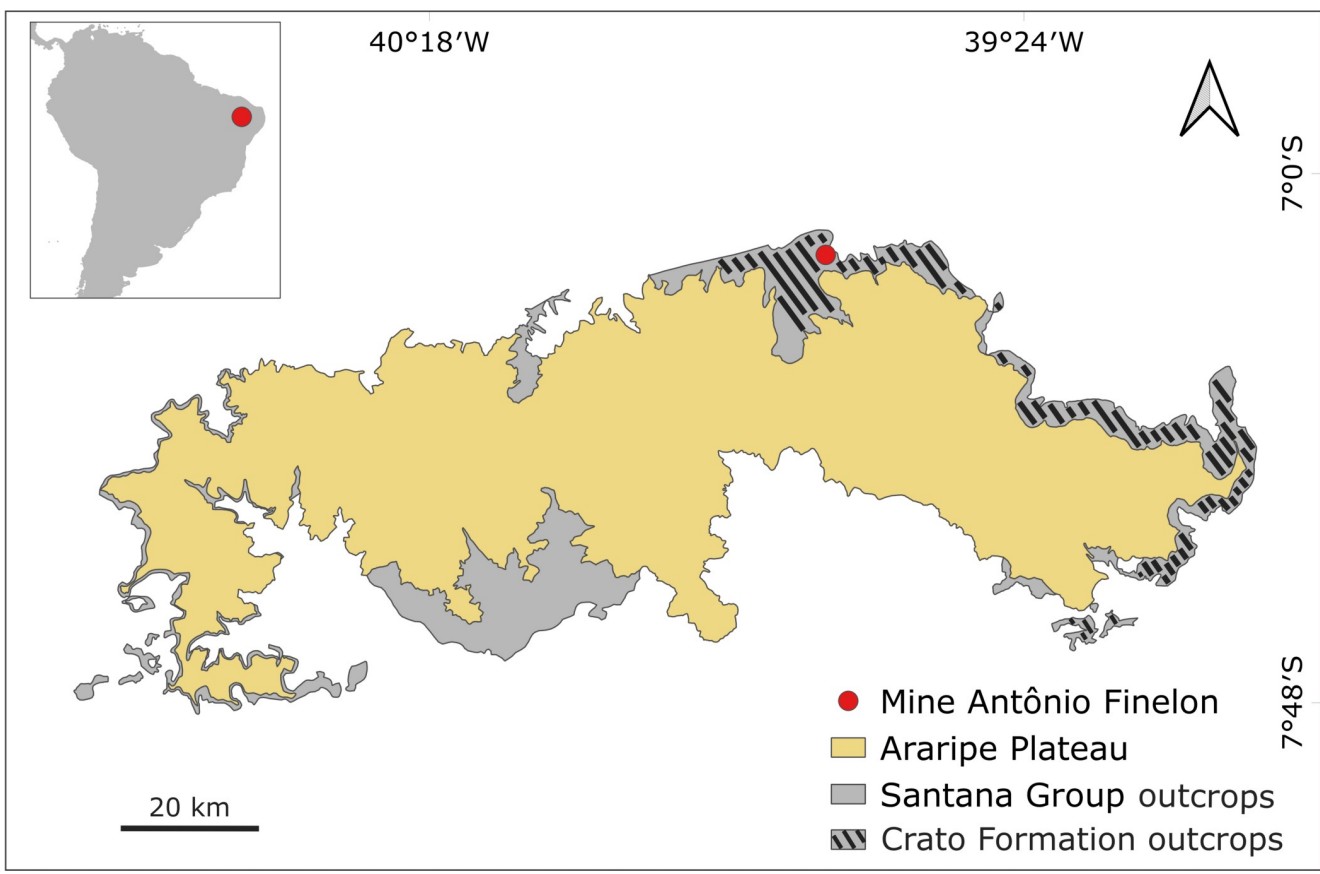

**Fig 1. Locality map.** Locality map showing the Mine Antônio Finelon in Nova Olinda municipality, where LPU 1696 was collected. The Araripe Plateau and outcrops of the Santana Group and the Crato Formation are also indicated.

and support compared to the equal weights scheme [14]. To estimate support of nodes, the Relative Bremer Support (RB) was calculated. All branches were collapsed with 15 additional steps than the shortest tree, generating 3,263 trees. The nodes in which the RB value was higher than 60 were considered strongly supported; between 30 and 60 were considered moderately supported; and those less than 30, poorly supported [15]. Also, 1,000 bootstrap (BT) pseudoreplicates were run, using the implicit enumeration search.

On average, 18% of characters were missing among OTUs represented by extant taxa, and 84% in the two extinct species. For this reason, we chose not to include data of the *Colocrus* larva in the analysis of the combined data matrix. Since the taxon only possesses seven characters to score in the matrix, it added a large amount of missing data that made many taxa unstable within branches, which could compromise the reliability of the results [16].

In addition to the analysis using the entire matrix, another one was performed using only larval characters (45 binary characters) to elucidate the phylogenetic position of *Colocrus indivicum* (once the associated adults are no longer considered conspecific, see below). The analysis followed the same protocol as detailed above, except no branch supports were estimated.

## Nomenclatural acts

The electronic edition of this article conforms to the requirements of the amended International Code of Zoological Nomenclature, and hence the new names contained herein are available under that Code from the electronic edition of this article. This published work and the nomenclatural

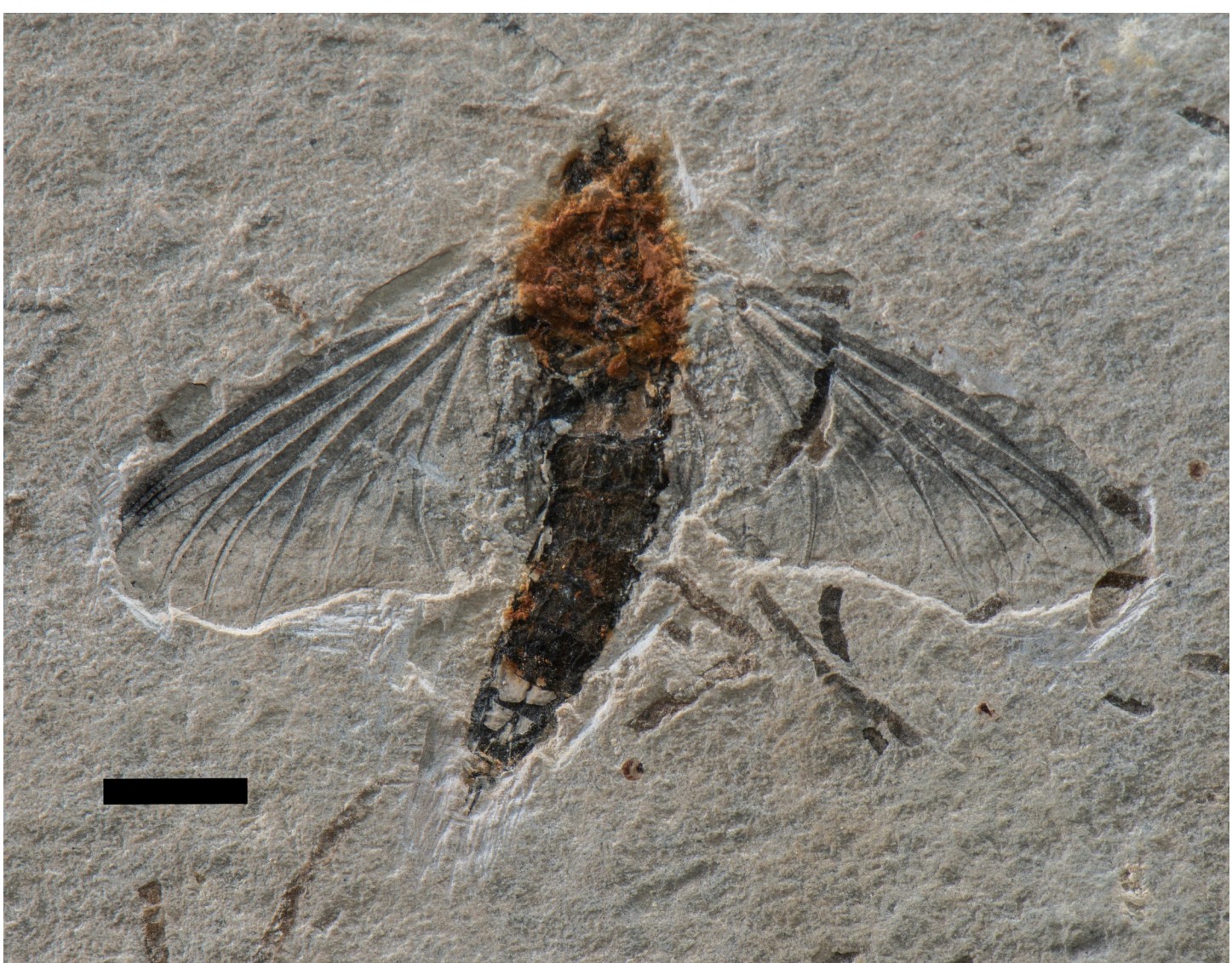

**Fig 2.** *Incogemina nubila* **gen. et sp. nov. holotype LPU 1696, adult.** Upper Aptian, Lower Cretaceous, Crato Formation, Araripe Basin. Nova Olinda municipality, Ceará State, Brazil. Photograph in dorsal view. Scale bar 5 mm.

acts it contains have been registered in ZooBank, the online registration system for the ICZN. The ZooBank LSIDs (Life Science Identifiers) can be resolved and the associated information viewed through any standard web browser by appending the LSID to the prefix "http://zoobank.org/". The LSID for this publication is: urn:lsid:zoobank.org:pub:EFE79A8D-18A8-4487-9748-2B80F4C F52B3. The electronic edition of this work was published in a journal with an ISSN, and has been archived and is available from the following digital repositories: PubMed Central, LOCKSS.

## Results

### Phylogenetic analysis

Parsimony analysis under implied weights for the combined data matrix of larval and adult characters resulted in two most parsimonious trees. The values of total fit and adjusted homoplasy were 57.18571 and 14.81429, respectively. The strict consensus tree is shown in Fig 4.

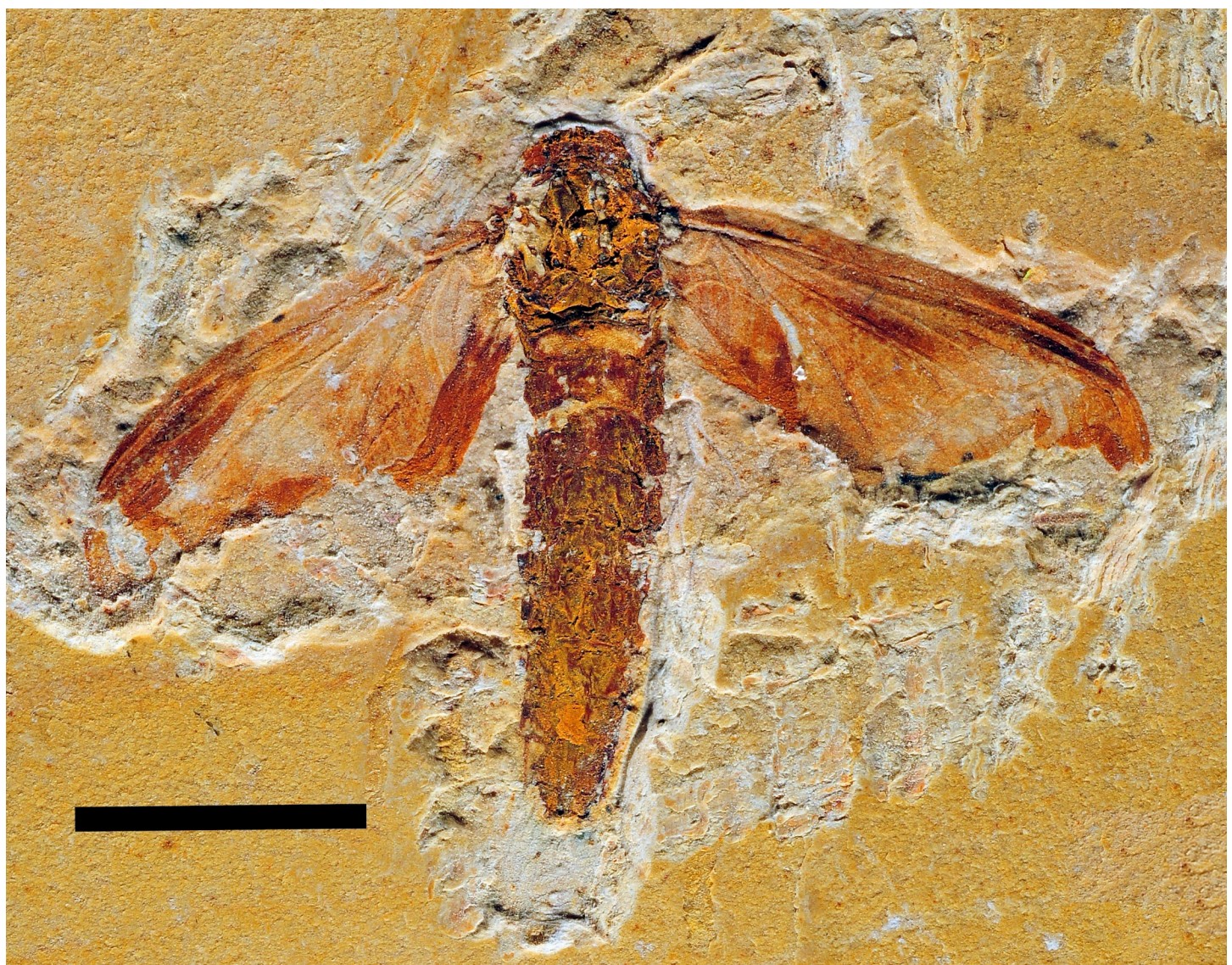

**Fig 3. *Colocrus indivicum.*** Paratype AMNH 43499, adult. Upper Aptian, Lower Cretaceous, Crato Formation, Araripe Basin. Ceará State, Brazil. Photograph in dorsal view. Scale bar 5 mm.

Oligoneuriidae was recovered as monophyletic, with BT and RB values of 80 and 61 respectively, supported by six synapomorphies: apical angle of maxilla acute (7:0); second segment of maxillary palp much longer in relation to first (8:1); lateral margin of epimera and episterna projected laterally (10:1); outer margin of fore femur without fine, long, simple setae (14:0); both adult tarsal claws rounded (50:1); vein RP of forewing originating next to wing base (59:1). *Chromarcys* was recovered as the sister-group to all other oligoneuriids and was supported by two apomorphies: terminal filament with setae, except at the basal 1/3 (43:1); forceps with three or more apical segments (66:1). LPU 1696 + Oligoneuriinae were recovered as a monophyletic group, with BT and RB values of 82 and 44 respectively, supported by one synapomorphy: presence of longitudinal vein gemination (pairing of longitudinal veins) (55:1). LPU 1696 was recovered with one autapomorphy: presence of incomplete gemination (76:0). Oligoneuriinae was recovered as monophyletic, with BT and RB values of 96 and 31

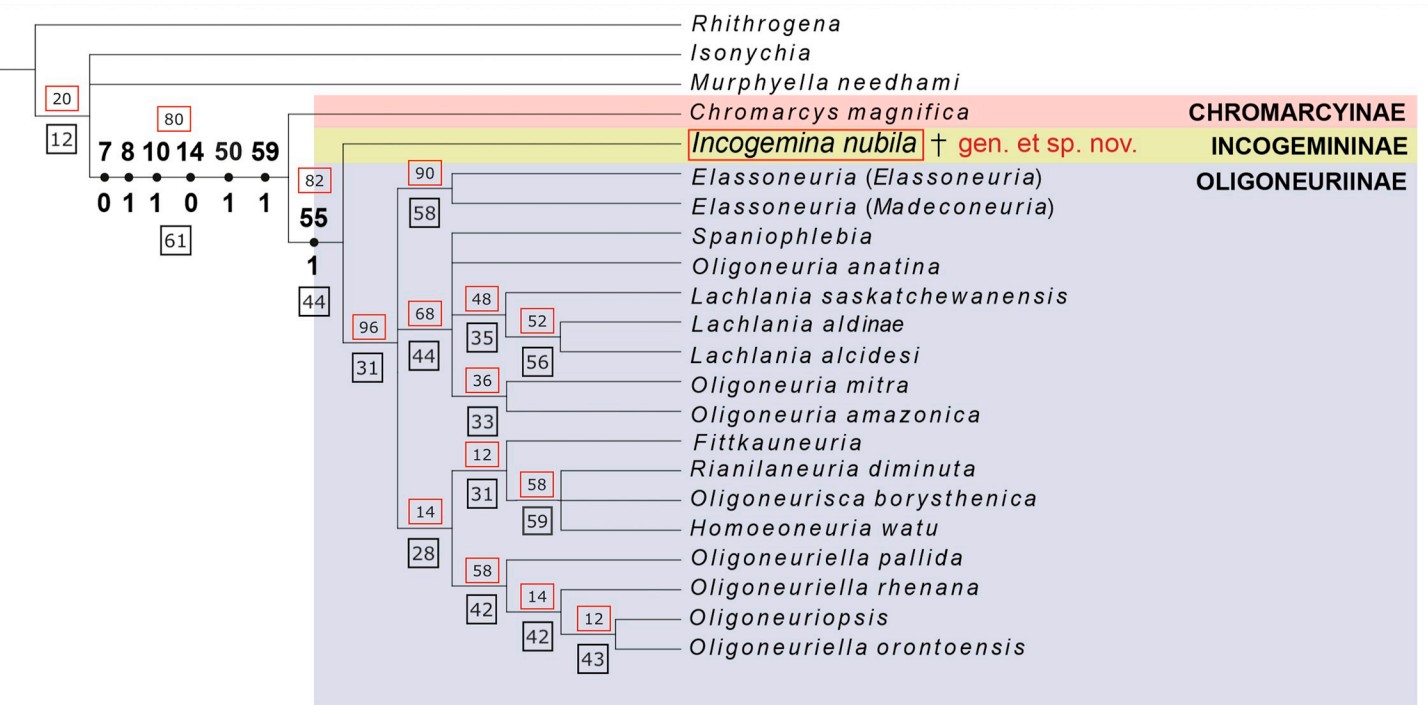

**Fig 4. Strict consensus tree.** Strict consensus of the two most parsimonious trees, from the analysis of 76 morphological characters from larvae and adults of Oligoneuriidae. Numbers above and below circles correspond to synapomorphies supporting Oligoneuriidae and LPU 1696 + Oligoneuriinae (characters and states, respectively). Other apomorphies are omitted for clarity. Bootstrap (BT) values are indicated above branches, in red squares, and Relative Bremer (RB) support values are located below branches, in black squares.

respectively, supported by three synapomorphies already found by Massariol et al. [4], plus the absence of cubital intercalaries (74:1; new character). Relationships between its genera were not well resolved, with many taxa in polytomies. Analysis of the matrix containing only larval characters was affected by the lack of adult characters, with a large politomy at the base of Oligoneuriidae, and relationships among several genera were spurious: *Isonychia* was recovered within Oligoneuriidae, while *Colocrus* and *Chromarcys* were recorded as sister-groups within Oligoneuriinae.

## Systematic paleontology

Subphylum Hexapoda Latreille, 1825
Class Insecta Linnaeus, 1758
Order Ephemeroptera Hyatt & Arms, 1890
Family Oligoneuriidae Ulmer, 1914
Subfamily Incogemininae nov.
urn:lsid:zoobank.org:act:377214A2-FACD-48C7-9E55-31C8972AB4E4
**Type genus.** *Incogemina* gen. nov.
**Branch-based definition.** All species more closely related to *Incogemina* than to *Oligoneuria* and *Chromarcys*.
**Diagnosis.** That of genus *Incogemina* gen. nov., as Incogemininae is monogeneric.
Genus *Incogemina* gen. nov.
urn:lsid:zoobank.org:act:C598DE19-4A9C-43B1-961A-B0E3E31DA24A
**Type species.** *Incogemina nubila* sp. nov.

**Derivation of name.** Named after the presence of incomplete gemination, from the Latin prefix *incohatus* combined with *geminae*.

**Diagnosis.** Forewing with crossveins distributed in all sectors; presence of incomplete gemination in longitudinal veins; presence of intercalary vein between $MP_1$ and $MP_2$; $MP_2$ and CuA running closely parallel for entire length; presence of intercalaries in cubital region.

*Incogemina nubila* sp. nov.

urn:lsid:zoobank.org:act:68095038-5398-4DF5-89A4-88E62EF89D8F

**Holotype.** Specimen no. LPU 1696, at the Paleontology Collection of the Regional University of Cariri (URCA), Crato, CE–Brazil (Fig 2).

**Type locality.** Mine Antônio Finelon, Nova Olinda municipality, Ceará state, Brazil.

**Referred specimen.** SMNS 66623, at the Staatliches Museum für Naturkunde Stuttgart, Germany. Exact locality unknown.

**Locality and horizon.** Southern Ceará state, Brazil. Upper Aptian, Lower Cretaceous [8], Crato Formation, Santana Group, Araripe Basin.

**Derivation of name.** Named after its grayish wing coloration.

**Diagnosis.** That of genus *Incogemina* gen. nov., monotypic.

**Description.** Specimen preserved in dorsal view, with both forewings articulated and spread out. Head and thorax hard to describe due to incomplete preservation. Hind wings, legs, antennae and most of the caudal filaments missing (Fig 2).

Body length: 23mm. Forewing length: 18 mm; subtriangular; ratio of wing length to width about 2:1; crossveins present in entire wing (weak in basal sections) (Fig 5); Sc and RA running parallel to each other; Sc and RA reaching wing apex; RP forks basally at one-fifth of wing length; $RP_1$ and IRP branch symmetrical at about one-quarter of distance from wing base; $RP_2$ branch at about one-third distance from wing apex, parallel at apex to IRP; $RP_{3-4}$ running parallel to $MA_1$; MA fork symmetrical, distal to midlength; MP fork near base; pre-gemination of $MP_1$ with $MA_2$ distally; two intercalaries between $MP_1$ and $MP_2$; $MP_2$ and CuA curving posteriorly forming a close parallel pair for entire length; $CuA_1$ curving posteriorly near tornus of wing; CuA branched forming $CuA_1$ and $CuA_2$; $CuA_2$ running towards posterior wing margin; CuP not branched, curved (Fig 5); anal veins difficult to trace. Abdomen with nine discernible segments; caudal filaments missing, with only the base of the two cerci preserved, representing the 10th segment.

**Comments.** Staniczek [3] mentioned a putative adult for *Colocrus magnum*. This specimen (SMNS 66623) was compared with *Incogemina* by photographs and is here referred to *Incogemina nubila*.

As Kluge [10] pointed out, "Mayfly systematics is based on a combination of larval, subimaginal and imaginal characters; however, larvae and winged stages (subimago and imago) are so different, that their association represents a special problem". There are several known instances of associations between larvae and adults that were later proved to be mistaken [10]. Even when dealing with extant individuals, it is hard to associate larvae and imagoes based only on morphological characters, and in most situations a precise identification can be made only by rearing them [10] or using DNA tools such as barcoding [17, 18]. There are oligoneuriid larval individuals described for the Crato Formation (the holotypes of *Colocrus indivicum* McCafferty, 1990—AMNH 43484 and *Colocrus magnum* Staniczek, 2007—SMNS 66624). The holotype of *Colocrus magnum* has a similar size to our specimen (LPU 1696), but other than that, is not comparable, so the association of winged and larval individuals is doubtful. Therefore, we prefer to describe winged stages under separate names, as is usually accepted in paleoentomology [19, 20].

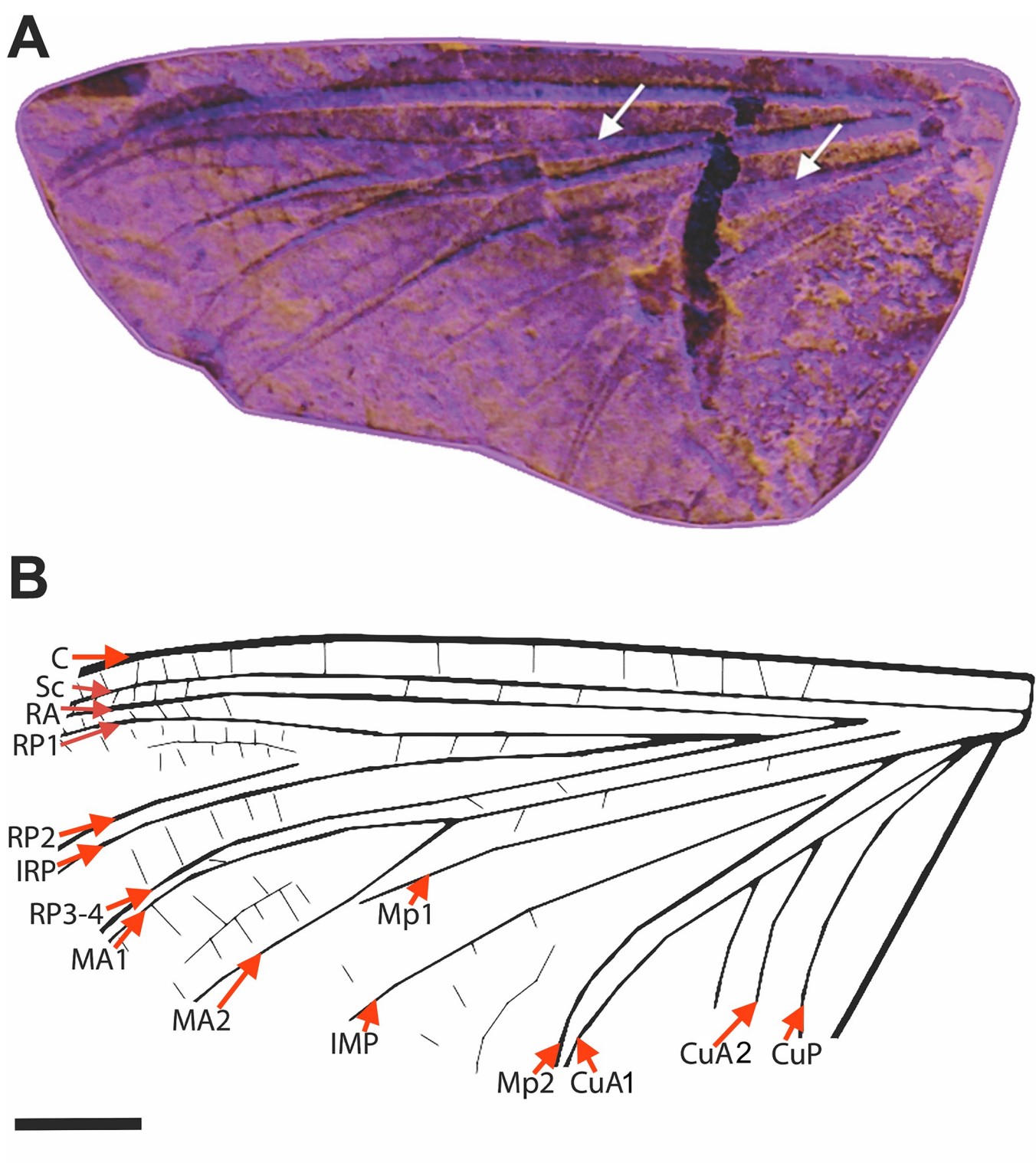

**Fig 5. *Incogemina nubila* gen. et sp. nov. holotype LPU 1696, adult.** Upper Aptian, Lower Cretaceous, Crato Formation, Araripe Basin. Nova Olinda municipality, Ceará State, Brazil. A) Right forewing, photograph under ultraviolet light evidencing the crossvenation. White arrows point to crossvenation. B) Left forewing, base venation, interpretative drawing. Scale bars = 2 mm.

### Comments on *Colocrus indivicum* McCafferty, 1990

McCafferty [2] described *Colocrus indivicum* based on two specimens, one larva (holotype) and one adult (paratype), and placed the species in the family Oligoneuriidae. The paratype, AMNH 43499, was preserved with some parts of the forewing folded, giving the impression of some pre-geminated veins. However, we here revise its venational data, which clearly indicate its placement in the family Hexagenitidae, because the CuA branches with triads between $CuA_1$ and $CuA_2$ (Fig 6). In fact, the only ephemeropterans that present such branching are the Hexagenitidae, among extinct and extant taxa. Furthermore, the relatively large hind wings are typical of the Hexagenitidae [3]. Hexagenitids, widespread during the Jurassic and Early Cretaceous, were often of large size, but the Lower Cretaceous taxa were moderate in size [21]. Hexagenitidae is the most common ephemeropteran family found in the Crato Formation [22, 23], but a more detailed taxonomic review of this clade is beyond the scope of the present work.

The species *Colocrus indivicum*, therefore, does not possess any adult representative. Here, we revise the diagnosis suggested by Massariol et al. [4] for the subfamily Colocrurinae: excluding all adult characters, only one larval character is left (abdominal gill I inserted dorsally). As this character state is present in most extant Ephemeroptera including the outgroups herein analyzed (see Massariol et al. [4]), it is actually a plesiomorphy, and thus not diagnostic. More complete specimens and detailed anatomical descriptions are thus needed to better define *Colocrus* and Colocrurinae. Modern imaging techniques, such as CT scans, may help uncover features currently hidden within the limestone.

## Discussion

Demoulin [24] emphasized similarities between *Chromarcys* and the Hexagenitidae, assuming a closer phylogenetic relationship between these taxa, which may explain McCafferty's [2]

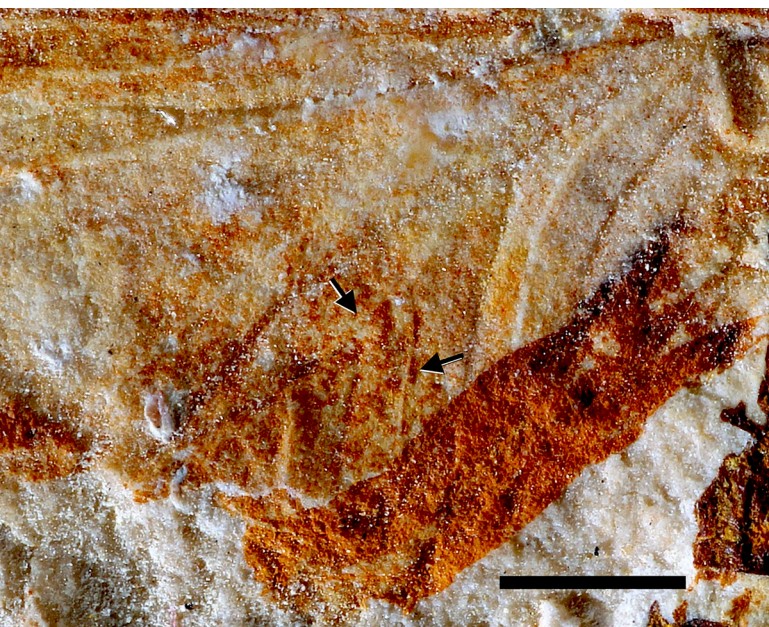

**Fig 6. *Colocrus indivicum.*** Paratype AMNH 43499, adult. Upper Aptian, Lower Cretaceous, Crato Formation, Araripe Basin. Ceará State, Brazil. Photograph of left forewing. Black arrows point to triads between $CuA_1$ and $CuA_2$. Scale bar 1 mm.

dubious placement of the adult specimen (paratype) of *Colocrus indivicum* as a representative of Oligoneuriidae. However, the Hexagenitidae clearly lack the apomorphies of the Oligoneuriidae.

All analyses published so far recovered Oligoneuriidae as monophyletic [4, 25]. Ogden and Whiting [26] and Ogden et al. [27] took in few Oligoneuriidae taxa, and their analyses also indicated the monophyly of the family. Following the criteria proposed by Hillis and Bull [15] for RB values, Oligoneuriidae was recovered in our analysis as strongly supported, with *Incogemina* + Oligoneuriinae and Oligoneuriinae as moderately supported.

*Incogemina* exhibits numerous venational similarities with *Chromarcys*, and differing from the Oligoneuriinae, such as forewings with numerous crossveins, some longitudinal veins of forewing not geminated, R2-5 branching toward basal fifth of forewing, and cubital sector of forewing developed (Fig 5). Our analysis demonstrates that these characters are plesiomorphic for Oligoneuriidae, and are shared within the remaining Heptagenioidea.

The venation of *Incogemina* appears to be a mosaic between a plesiomorphic ephemeropteran wing venation, as evident in *Chromarcys*, and the highly reduced apomorphic wing with geminated longitudinal veins that is found in Oligoneuriinae. Major intercalaries except for IMP are absent in *Incogemina* as they are in Oligoneuriinae, but part of the radial and cubital area retain remnants of the plesiomorphic ephemeropteran venation, as in *Chromarcys*. A tendency towards gemination is clearly evident in *Incogemina*, and is the most important evidence that this species bridges a gap in morphology between the Oligoneuriinae and other Oligoneuriidae.

The discovery of a Gondwanan species with such a combination of characters is expected. It appears that several families of Ephemeroptera have had their primary evolutionary development on the Gondwana supercontinent [28]. Subsequently these southern groups have either dispersed to the temperate areas of the northern Hemisphere [29] or suffered vicariance [4]. A Gondwanan origin of the family Oligoneuriidae was suggested by Edmunds [28, 29], McCafferty [30], and more recently by Massariol et al. [4].

Massariol et al. [4] proposed that the divergence between Oligoneuriinae and Chromarcyinae was related to the breakup of Gondwana. The initial evolution of oligoneuriids in Gondwana was complex, and clades such as *Colocrus* and *Incogemina* might have been sympatric, although more refined stratigraphic data is missing. Reasons for the divergence between major clades are unclear, but our findings give support to the hypothesis that the divergence between Oligoneuriinae and Incogemininae probably occurred in South America and that the present biogeographical distribution of *Chromarcys* can be explained by dispersal and later by vicariance, as commented by Massariol et al. [4]. The finding of additional and well-preserved fossilized specimens may test this observation.

## Conclusions

*Incogemina nubila* gen. et sp. nov. constitutes the second known fossilized adult of an oligoneuriid, and the first specimen to be described in detail, adding to current knowledge of the mayfly diversity in the Lower Cretaceous. The discovery of a new subfamily of Oligoneuriidae in the Mesozoic of South America fills important gaps in the evolutionary history of the family. From a morphological point of view, the specimens of this new taxon fill gaps between an oligoneuriid ancestor and the extant *Chromarcys*, because they present a phenotype of wing venation that combines plesiomorphic and apomorphic character states. From a biogeographical view, they demonstrate that the divergence between Oligoneuriinae and Incogemininae probably occurred in what is now South America.

## Supporting information

**S1 Appendix. Morphological characters.** Morphological characters and their states coded for this phylogenetic analysis (from Massariol *et al.*, [4], except where noticed).
(DOCX)

**S2 Appendix. Character matrix.** Matrix of morphological characters and states used for the phylogenetic analyses of Oligoneuriidae. Ready for use in TNT format.
(DOCX)

## Acknowledgments

The authors thank the editor Martha Richter, and Fabiana Massariol and an anonymous reviewer, for their helpful contributions. We also thank Janice Peters and Arnold Staniczek for their valuable observations in a previous version of the manuscript. We are grateful to David Grimaldi and Courtney Richenbacher (AMNH) for providing pictures of specimen AMNH 43499. We thank Renan Bantim, who received us at the Laboratório de Paleontologia of URCA, and Lucio Silva for receiving us at the Museu de Paleontologia "Plácido Cidade Nuvens" in Santana do Cariri. We also thank Marcelo Tavares and Roman Godunko for useful comments that improved the article. APS thanks Edú Guerra and Richard Buchmann for constructive assistance on image editing. TNT is freely available thanks to a subsidy from the Willi Hennig Society.

## Author Contributions

**Conceptualization:** Arianny P. Storari, Taissa Rodrigues, Antonio A. F. Saraiva, Frederico F. Salles.

**Data curation:** Antonio A. F. Saraiva.

**Investigation:** Arianny P. Storari.

**Project administration:** Arianny P. Storari, Taissa Rodrigues.

**Resources:** Taissa Rodrigues, Antonio A. F. Saraiva.

**Supervision:** Taissa Rodrigues, Frederico F. Salles.

**Validation:** Arianny P. Storari, Frederico F. Salles.

**Writing – original draft:** Arianny P. Storari.

**Writing – review & editing:** Arianny P. Storari, Taissa Rodrigues, Antonio A. F. Saraiva, Frederico F. Salles.

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
