## [Decision Letter · Decision Letter 0]

8 Sep 2020

PONE-D-20-24186

Unmasking a gap: new fossil oligoneuriid (Ephemeroptera: Insecta) from the Crato Formation (upper Aptian), Araripe Basin, NE Brazil, with comments on Colocrus McCafferty

PLOS ONE

Dear Dr. Storari,

Thank you for submitting your manuscript to PLOS ONE. After careful consideration, we feel that it has merit but does not fully meet PLOS ONE’s publication criteria as it currently stands. Therefore, we invite you to submit a revised version of the manuscript that addresses the points raised during the review process.

We look forward to receiving your revised manuscript.

Kind regards,

Martha Richter, PhD

Academic Editor

PLOS ONE

2. We note that Figure 1 in your submission contain map images which may be copyrighted. All PLOS content is published under the Creative Commons Attribution License (CC BY 4.0), which means that the manuscript, images, and Supporting Information files will be freely available online, and any third party is permitted to access, download, copy, distribute, and use these materials in any way, even commercially, with proper attribution. For these reasons, we cannot publish previously copyrighted maps or satellite images created using proprietary data, such as Google software (Google Maps, Street View, and Earth). For more information, see our copyright guidelines: http://journals.plos.org/plosone/s/licenses-and-copyright.

2.1.    You may seek permission from the original copyright holder of Figure 1 to publish the content specifically under the CC BY 4.0 license. 

2.2.    If you are unable to obtain permission from the original copyright holder to publish these figures under the CC BY 4.0 license or if the copyright holder’s requirements are incompatible with the CC BY 4.0 license, please either i) remove the figure or ii) supply a replacement figure that complies with the CC BY 4.0 license. Please check copyright information on all replacement figures and update the figure caption with source information. If applicable, please specify in the figure caption text when a figure is similar but not identical to the original image and is therefore for illustrative purposes only.

Reviewers' comments:

Reviewer's Responses to Questions

**Comments to the Author**

1. Is the manuscript technically sound, and do the data support the conclusions?

Reviewer #1: Yes

Reviewer #2: Yes

2. Has the statistical analysis been performed appropriately and rigorously? 

Reviewer #1: Yes

Reviewer #2: Yes

3. Have the authors made all data underlying the findings in their manuscript fully available?

Reviewer #1: Yes

Reviewer #2: Yes

4. Is the manuscript presented in an intelligible fashion and written in standard English?

Reviewer #1: Yes

Reviewer #2: Yes

5. Review Comments to the Author

Reviewer #1: This manuscript by Storari et al. is a clearly written and pleasantly concise paper that includes a description and phylogenetic contextualization of a new fossil ephemeropteran genus. The authors’ conclusions are well stated and supported by the data presented here and summarized from previous publications. The figures and text cover all the essentials. I recommend very minor edits below, but feel that this paper could almost be published as-is.

Minor suggestions:

Recommend adding dagger symbol (†) to fossil taxa on the phylogeny.

Please add bootstrap values to the phylogeny – if numbers would clutter too much, perhaps consider a color scheme red/yellow/green for certain bootstrap ranges. At the moment, only a few support values are reported in the text.

Line 20: “mayflies stand out” – why do they stand out? Morphologically conspicuous?

Line 37: if you have quantitative information on the specimen-level disparity between larvae and adults from the McCafferty paper, could be worth including (ie 10:1 ratio, or similar)

Line 47: “Indomalayan” is more commonly used in place of “Oriental Region”

Line 89: although not required, it’s often helpful in morphological matrices that include fossil taxa to report the number of missing character states. I recommend including the total % of missing cells in the matrix, the average number of cells per taxon, and the number of missing cells for fossils/new taxon. These quick stats will help readers assess how missing data may be informing relationships.

Line 94: “nonaddictive” should be “non-additive” I believe

Line 389: unclear what “hendata” is

Reviewer #2: I am very glad the authors wrote this manuscript, describing in detail the new fossil of Oligoneuriidae, proposing a new subfamily, as well as its phylogenetic positioning within the family. They also advanced in the knowledge of other known Oligoneuriidae fossils (Colocrus indivicum and Colocrus magnum), besides raising biogeographical hypotheses.

The manuscript is well-written and with a good balance among descriptions, illustrations, and discussions. Just minor revisions are necessary to clarify some details. Thus, I suggest the publication of the present work after making these small revisions:

1) Line 39. You could specify that the species refer to Ephemeroptera. As it is written it can be confused with Oligoneuriidae species;

2) Line 49. I strongly suggest you include the information "gender unknown";

3) Lines 73, 80, 223, 234. In original description of Colocrus indivicum (McCafferty 1990), the paratype code is: AMNH 43499. I believe that a kind of typo must have occurred;

4) Lines 85, 86. Just for reasons of order, I suggest that the information be inverted, mentioning character 32 first and then 54;

5) Line 89. 21 ingroup taxa (OTU) were used in Massariol et al. (2019), so you excluded from yours analyses Lachlania aldinae, Lachlania sp., and Homoeoneuria sp. Please explain the reason for this exclusion or make it clear that you have not used exactly all OTUs from Massariol et al. (2019);

6) Line 94. Why didn't you do an exhaustive search (implicit enumaration command), since the matrix is small? I think it is worth doing this analysis, as this will explore the total number of possible trees for the data matrix;

7) Line 96. Please explain the choice of k value;

8) Line 98. Why did you choose bootstrap to calculate branch support? For parsimony analyzes there are alternatives with less bias, such as Relative Bremer support (RB) or frequency difference (GC). For more details see: Goloboff PA, Farris JS, Kallersjo M, Oxelman B, Ramirez MJ, Szumik CA. 2003. Improvements to resampling measures of group support. Cladistics 19: 324–332;

9) Line 100. Please specify "combined data". Later you explain that it is the matrix with larvae and adults characters, but I think it is better to make this information clear at "Material and Methods" topic;

10) Line 101. I counted 7 characters instead of 5: 27, 28, 30, 32, 33, 34 and 37. Please check it again;

11) Line 124. I suggest you include the values of "total fit" and "adjusted homoplasy";

12) Line 128. For a clearer figure caption, I suggest you include the information that characters are from larvae and adults;

13) Line 129. Not all synapomorphies were depicted in Fig. 4. Explain which ones have been suppressed and why. In addition, I did the analysis again in TNT and at Oligoneuriidae clade, another synapomorphy was recovered (in addition to the ones you found): 14 (0). Review the analysis;

14) Please check if there are 5 or 6 (see comment 13);

15) Lines 133-144. If you prefer, you can simplify by “8:1”, “10:1” and so on instead of using “character 8: state 1” and “character 10: state 1”;

16) Line 146. "Inconclusive" is a vague term, so I suggest you explain the results obtained. Did Colocrus go into polytomy? Was there no high support from the branches?;

17) Line 157. In the sentence did you really mean Oligoneuria and not Oligoneurinae?

18) Lines 191, 192. I also suggest you pointing out the CuA1 and CuA2 veins in figure 5B;

19) Lines 282, 295. Did you mean the divergence between Oligoneuriinae and Incogemininae or the divergence between Oligoneuriinae + Incogemininae and other Oligoneuriidae? If it is the first option, replace "+" with "-", so the idea will be clearer.

20) Line 283. The current distribution of Chromarcys is explained both by dispersion, at ancient moment, and by vicariance later. Review this;

21) Line 284. Did you mean fossil or extant specimens? Please clarify this information.

6. PLOS authors have the option to publish the peer review history of their article (what does this mean?). If published, this will include your full peer review and any attached files.

Reviewer #1: No

Reviewer #2: **Yes: **Fabiana Criste Massariol

---

## [Author Response · Author response to Decision Letter 0]

21 Sep 2020

Concerning the considerations made by the Editor, we substituted Figure 1, that was made using GIS, since the previous map was modified from Barling et al. (2015).

Reviewer 1 recommended changes in the phylogeny image (Figure 3), such as the addition of a dagger symbol to indicate taxa known only by fossils and addition of support values to all branches, besides minor modifications of the text. We complied with all changes.

Concerning Reviewer 2, most of the suggestions were related to the phylogenetic analysis, such as to perform an exhaustive search instead of a traditional search and to calculate branch support using Relative Bremer. These modifications were made. Minor modifications of the text were also made after the reviewer’s suggestion. 

Detailed replies to reviewers’ comments were addressed in the document 'Response to Reviewers'.

Thank you for handling this manuscript.

---

## [Editor Report · Decision Letter 1]

25 Sep 2020

Unmasking a gap: a new oligoneuriid fossil (Ephemeroptera: Insecta) from the Crato Formation (upper Aptian), Araripe Basin, NE Brazil, with comments on Colocrus McCafferty

PONE-D-20-24186R1

Dear Dr. Storari,

We’re pleased to inform you that your manuscript has been judged scientifically suitable for publication and will be formally accepted for publication once it meets all outstanding technical requirements.

Kind regards,

Martha Richter, PhD

Academic Editor

PLOS ONE

---

## [Editor Report · Acceptance letter]

5 Oct 2020

PONE-D-20-24186R1 

Unmasking a gap: a new oligoneuriid fossil (Ephemeroptera: Insecta) from the Crato Formation (upper Aptian), Araripe Basin, NE Brazil, with comments on *Colocrus* McCafferty 

Dear Dr. Storari:

I'm pleased to inform you that your manuscript has been deemed suitable for publication in PLOS ONE. Congratulations! Your manuscript is now with our production department. 

Kind regards, 

on behalf of

Dr. Martha Richter 

Academic Editor

PLOS ONE